# Iterative Connecting Probability Estimation for Networks

**Yichen Qin**
University of Cincinnati
qinyn@ucmail.uc.edu

**Linhan Yu**
Renmin University of China
yulinhan47@foxmail.com

**Yang Li**[*]
Renmin University of China
yang.li@ruc.edu.cn

## Abstract

Estimating the probabilities of connections between vertices in a random network using an observed adjacency matrix is an important task for network data analysis. Many existing estimation methods are based on certain assumptions on network structure, which limit their applicability in practice. Without making strong assumptions, we develop an iterative connecting probability estimation method based on neighborhood averaging. Starting at a random initial point or an existing estimate, our method iteratively updates the pairwise vertex distances, the sets of similar vertices, and connecting probabilities to improve the precision of the estimate. We propose a two-stage neighborhood selection procedure to achieve the trade-off between smoothness of the estimate and the ability to discover local structure. The tuning parameters can be selected by cross-validation. We establish desirable theoretical properties for our method, and further justify its superior performance by comparing with existing methods in simulation and real data analysis.

## 1 Introduction

Network analysis has been a promising research area in data science and widely applied in many fields, including medicine, social sciences. Objects such as genes, proteins, or people are presented as vertices in the network, and relationships between objects (e.g., protein interactions or friendships) are presented as connections between pairs of vertices. In this paper, we focus on estimating the connecting probability between each pair of vertices based on the observed network.

In the literature, the most fundamental assumption is the Aldous-Hoover theorem (Aldous, 1981). Assume that the network is vertex-exchangeable, then each vertex $i$ is associated with a latent variable $\xi_i$, and the connecting probability between any two vertices depends on a graphon function $f(\cdot)$ of the corresponding latent variables.

To estimate the connecting probabilities, some researchers focus on estimating $f(\cdot)$ under strong assumptions. Chan and Airoldi (2014) propose the sorting-and-smoothing (SAS) algorithm. It first sorts the adjacency matrix using the empirical degrees and smoothes the sorted matrix via total variation minimization. It requires strict monotonicity of the degrees, which is difficult to be satisfied in practice. Since the purpose of estimating $f(\cdot)$ is to estimate the connecting probabilities, another direction is to estimate the latter directly, such as stochastic block models approximation (SBA) (Airoldi et al., 2013; Holland et al., 1983; Wang and Wong, 1987). Under the SBM, vertices can be divided into several blocks, and the connecting probability between two vertices depends only on the blocks they belong to. Once the vertices are clustered correctly, the connecting probabilities can be estimated by a moment estimator. In recent years, methods based on SBM approximation and its variation, network histograms, have been widely studied (Choi and Wolfe, 2014; Olhede

---

[*]Correspondence Author, Center for Applied Statistics and School of Statistics, Renmin University of China

35th Conference on Neural Information Processing Systems (NeurIPS 2021).

and Wolfe, 2014). Gao and Ma (2021) establish the framework of minimax error rate analysis for these methods. Since the adjacency matrix can be viewed as a noisy version of the connecting probability matrix, general matrix denoising methods can be applied too. Chatterjee (2015) provides a universal singular value thresholding method (USVT), and Xu (2018) analyzes its error rate. Zhang et al. (2017) propose an efficient neighborhood smoothing method (NS). They select those with similar connecting patterns as the neighbors for each vertex, then the connecting probabilities can be estimated by neighborhood averaging. However, the estimate is not reused to improve neighborhood selection, which may lead to information loss.

Since the main purpose of estimating the graphon function is to obtain the connecting probabilities, we focus on estimating the connecting probability matrix and develop an iterative connecting probability estimation method (ICE) in this paper. We define a new vertex distance calculated directly on the probability matrix. In contrast to one-step neighborhood smoothing method, our method is iterative. In each iteration, with an estimate, it can update the pairwise vertex distances and the sets of similar vertices. Then, with better selected similar vertices, it can improve the performance on estimation, making it easier to discover complicated structure. The initial input of the iterations can be an estimate obtained by the existing methods as we introduce above. Even if it begins with a random neighborhood averaging estimate, it can converge quickly and perform well in many cases. We also propose a two-stage strategy for neighborhood selection, which uses different sizes of the similar vertex sets during and at the end of the iterations, aiming to cope with the trade-off between the smoothness of the estimate and the ability to discover local structure. The tuning parameters can be selected by network cross-validation. According to the results on simulated and real networks, our method is comparable to state-of-the-art methods in simple cases and outperforms them in the presence of complicated structure.

## 2 Framework

Suppose $G = (V, E)$ is the observed undirected network where $V = [n]$ denotes the vertex set and $E$ denotes the edge set and $E \subseteq \{(i, j) : i, j \in V, i \neq j\}$. The network can be represented by an adjacency matrix $\mathbf{A} = [A_{ij}]^{n \times n} \in \{0, 1\}^{n \times n}$, where $A_{ij} = 1$ if vertices $i$ and $j$ are connected and 0 otherwise. We only consider undirected networks without self-loops, so $\mathbf{A}$ is symmetric with $A_{ii} = 0$ for $i \in V$. Assume that the network is vertex-exchangeable, according to the Aldous-Hoover theorem (Aldous, 1981), we have the following representation: for each vertex $i$, there exists a latent random variable $\xi_i \overset{i.i.d.}{\sim} \text{Uniform}(0, 1)$, then for each vertex pair $(i, j)$, we have $A_{ij} \sim \text{Bernoulli}(P_{ij})$, where $P_{ij} = f(\xi_i, \xi_j)$ and $f(\cdot)$ is called the graphon function. Let $\mathbf{P} = [P_{ij}]^{n \times n} \in [0, 1]^{n \times n}$ denote the probability matrix. Let $||\cdot||_2$ and $||\cdot||_F$ denote the $\ell_2$ and Frobenius norms respectively.

Given the observed adjacency matrix $\mathbf{A}$, our goal is to estimate the connecting probability matrix $\mathbf{P}$. Since for each pair $(i, j)$, $A_{ij}$ is the only realization of $P_{ij}$, it is hard to get a sample-based estimation like maximum likelihood estimation. However, if there exists another vertex $i'$ satisfying $f(\xi_{i'}, \cdot) \approx f(\xi_i, \cdot)$, then $A_{i'j}$ can be roughly viewed as another realization of $P_{ij}$ for the reason that $P_{ij} = f(\xi_i, \xi_j) \approx f(\xi_{i'}, \xi_j) = P_{i'j}$. Assume that there is such a group of vertices who share similar connecting patterns with vertex $i$; we refer to them in the rest of this paper simply as its "similar vertices". Let $S_i^*$ denote the set of similar vertices for vertex $i$, that is, $S_i^* = \{i' : \mathbf{P}_{i\cdot} \approx \mathbf{P}_{i'\cdot}\}$. Then we could estimate $P_{ij}$ by simply averaging the entries $A_{i'j'}$ over $i' \in S_i^*$ and $j' \in S_j^*$, that is,

$$\widetilde{P}_{ij} = \widetilde{P}_{ji} = (s^*)^{-2} \sum_{i' \in S_i^*} \sum_{j' \in S_j^*} A_{i'j'}, \tag{1}$$

where $s^* = |S_i^*|$ is the number of similar vertices. Ideally, with known $\mathbf{P}$, it is easy to define the set of similar vertices for vertex $i$ as $S_i^* = \{i' : 0 < d_{ii'}^* \leq d_{s^*}^*\}$, where $d_{ii'}^* = ||\mathbf{P}_{i\cdot} - \mathbf{P}_{i'\cdot}||_2^2/n$ denotes the distance between vertex $i$ and vertex $i'$ and $d_{s^*}^*$ is the $s^*$-th smallest elements of $\{d_{ii'}^* : i' \neq i\}$.

## 3 Methodology

### 3.1 Iterative Connecting Probability Estimation

The neighborhood averaging method in Section 2 is only conceptually feasible. In practice, without knowledge of $\mathbf{P}$, it is difficult to obtain the pairwise vertex distances and the sets of similar ver-

tices. We propose an iterative connecting probability estimation method (ICE) to solve the problem. Assume that we have an initial estimate $\widehat{\mathbf{P}}$, the estimated vertex distance $d_{ii'}$ can be formulated as $d_{ii'} = ||\widehat{\mathbf{P}}_{i\cdot} - \widehat{\mathbf{P}}_{i'\cdot}||_2^2/n$. Then we can estimate $S_i^*$ with $S_i = \{i' : 0 < d_{ii'} \leq d_s\}$, where $s = |S_i|$ is the estimated size of the similar vertex set and $d_s$ is the $s$-th smallest elements of $\{d_{ii'} : i' \neq i\}$. Then a new estimate can be obtained by neighborhood averaging with

$$\widehat{P}_{ij} = \widehat{P}_{ji} = s^{-2} \sum_{i' \in S_i} \sum_{j' \in S_j} A_{i'j'}. \tag{2}$$

Using such an estimate, we can update the pairwise vertex distances and the sets of similar vertices again, and obtain a new $\widehat{\mathbf{P}}$. We repeat the process above until $\widehat{\mathbf{P}}$ converges. Let $\mathbf{D} = [d_{ij}]^{n \times n}$ and $\mathbf{S} = \{S_1, \ldots, S_n\}$ denote the pairwise vertex distances and the sets of similar vertices respectively. The iterative procedure is inspired by the interdependence among $\mathbf{D}$, $\mathbf{S}$ and $\widehat{\mathbf{P}}$, as is shown in Figure 1. We outline the details of our method in Algorithm 1.

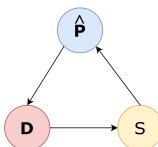

Figure 1: *The relationship among* $\mathbf{D}$, $\mathbf{S}$ *and* $\widehat{\mathbf{P}}$.

---

**Algorithm 1** Iterative connecting probability estimation method

---

**Input:** observed adjacency matrix $\mathbf{A}$; initial connecting probability estimate $\widehat{\mathbf{P}}^{(0)}$; neighborhood size $s$; threshold $\delta_0 > 0$.

**Output:** connecting probability estimate $\widehat{\mathbf{P}}$.

1: Let $\delta_{\mathbf{P}} = +\infty$ and $m = 0$.
2: **while** $\delta_{\mathbf{P}} > \delta_0$ **do**
3:      For each vertex pair $i, i' \in V$, obtain their distance $d_{ii'} = ||\widehat{\mathbf{P}}_{i\cdot}^{(m)} - \widehat{\mathbf{P}}_{i'\cdot}^{(m)}||_2^2/n$.
4:      For each vertex $i \in V$, obtain its set of similar vertices $S_i = \{i' : 0 < d_{ii'} \leq d_s\}$.
5:      For each vertex pair $i, j \in V$, update their connecting probability estimate $\widehat{P}_{ij}^{(m+1)} = s^{-2} \sum_{i' \in S_i} \sum_{j' \in S_j} A_{i'j'}$.
6:      Let $\delta_{\mathbf{P}} = ||\widehat{\mathbf{P}}^{(m+1)} - \widehat{\mathbf{P}}^{(m)}||_F / ||\widehat{\mathbf{P}}^{(m)}||_F$.
7:      Let $m = m + 1$.
8: **end while**
9: **return** $\widehat{\mathbf{P}} = \widehat{\mathbf{P}}^{(m)}$.

---

In Algorithm 1, an initial estimate $\widehat{\mathbf{P}}^{(0)}$ is required. Any $\widehat{\mathbf{P}}$ via existing methods can be used as $\widehat{\mathbf{P}}^{(0)}$. Alternatively, a random initial value would also work as discussed in Section 3.2, since our method is not sensitive to the initialization. For each vertex $i$, we get $S_i$ by randomly sampling $s$ vertices from $V \setminus \{i\}$ without replacement, then we can obtain such a random $\widehat{\mathbf{P}}^{(0)}$ by neighborhood averaging with these random selected vertices.

As to the size of the similar vertex set $s$, Zhang et al. (2017) set $s = C(n \log n)^{1/2}$ for each vertex $i$, where $C$ is recommended set as 1. In Section 3.3, we develop a two-stage procedure using different levels of $C$ in different stages, which improves the performance of our method significantly.

It should be noted that although we discuss our estimate as defined in (2) for convenience, in practice, in case that $S_i$ and $S_j$ may overlap, we modify it as

$$\widehat{P}_{ij} = \widehat{P}_{ji} = (s^2 - |S_i \cap S_j|)^{-1} \sum_{i' \in S_i, j' \in S_j, i' \neq j'} A_{i'j'}.$$

## 3.2 Iterative Estimation versus One-step Estimation

A main contribution of our work is to estimate $\mathbf{P}$ iteratively rather than output a one-step estimate. As for those one-step methods (Zhang et al., 2017), they obtain the set of similar vertices based on the distances calculated from $\mathbf{A}$, then obtain $\widehat{\mathbf{P}}$ by neighborhood averaging. However, if we update the set of similar vertices based on $\widehat{\mathbf{P}}$, the previous set of similar vertices may be changed. The

contradiction may cause inefficiency on making full use of the observed network. Indeed, if $\widehat{\mathbf{P}}$ is reliable, we should update the set of similar vertices to get a better estimate until the sets remain unchanged.

We compare the performances of our iterative estimation and one-step estimation with a simulated network which consists of 1000 vertices as an example. As shown in Figure 2(a), the $1000 \times 1000$ connecting probability matrix is full rank with complicated structure. We use root mean squared errors (RMSE) to evaluate the performance. We present the results of the 1st and the 10th iterations with two kinds of initial inputs and varying value of tuning parameter $C$ in Figure 2(b). It is clear that the iterations improve the precision of the estimates with both initializations.

The results also illustrate that our method is not sensitive to the initialization. Without iterations, the one-step estimate obtained by the NS method proposed by Zhang et al. (2017) outperforms random initialization significantly. However, after sufficient number of iterations, the performance of our method with different initial values becomes comparable. In practice, as the true network structure is unknown, we recommend using the random initial input.

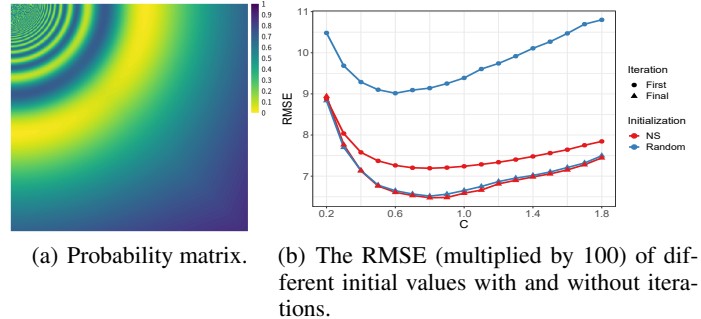

(a) Probability matrix.    (b) The RMSE (multiplied by 100) of different initial values with and without iterations.

Figure 2: *The probability matrix and the RMSE of ICE with and without iterations.*

### 3.3 Two-stage Neighborhood Selection

Setting an appropriate $C$ is an important task. A small $C$ is unable to make full use of the similar vertices while a large $C$ may lead to over-smoothing. To deal with the dilemma, we propose a two-stage strategy. In stage one, we use a small $C = C_{\mathrm{it}}$ during iterations to avoid over-smoothing and to discover local structure. In stage two, after the estimate converges, a large $C = C_{\mathrm{est}}$ is used to include more similar vertices to obtain a smooth estimate. The reason is that, as the iterations progress, the precision on neighborhood selection will be improved, so we should enlarge the size of the similar vertex set gradually. However, it is impracticable to select the upper bound, the lower bound and the step-size of $C$ simultaneously. So, we simplify the problem by setting two tuning parameters. The first tuning parameter helps us better order the vertices according to their latent positions, whereas the second tuning parameter helps us obtain better estimates using the correct number of vertices.

We compare our two-stage strategy and the strategy using a fixed $C$ ($C_{\mathrm{it}} = C_{\mathrm{est}}$) on the above simulated network. We consider the estimate of $\mathbf{P}_{i\cdot}$, where the index $i$ is set as $i = 350$ to present the local structure. As is shown in Figure 3(a), using a small fixed $C$ helps to obtain an estimate with small bias but introduces large variance. In contrast, according to Figure 3(b), by using a large fixed $C$, we obtain a smooth estimate but fail to capture the local structure. With the well-selected tuning parameters, the two-stage strategy outputs a smooth estimate of $\mathbf{P}_{i\cdot}$ while capturing the local structure well.

We present the RMSE of the estimates with different combinations of $(C_{\mathrm{it}}, C_{\mathrm{est}})$ on the whole $\mathbf{P}$ over 100 repetitions in Figure 3(c). The horizontal axis stands for the value of $C_{\mathrm{est}}$ used while different curves present the results with different $C_{\mathrm{it}}$. The curve of "Oracle" displays the RMSE when estimating $\mathbf{P}$ with the true similar vertices, which provides a golden standard. For all the curves, as $C_{\mathrm{est}}$ increases, RMSE decreases sharply first, reaches the bottom near $C_{\mathrm{est}} = 1$ and increases gradually, which illustrates that using an appropriate $C_{\mathrm{est}}$ is important. Comparing the different curves, it is easily seen that with a fixed $C_{\mathrm{est}}$, a smaller $C_{\mathrm{it}}$ will lead to smaller RMSE since it helps to discover local structure. Another idea to achieve the trade-off is to use a fixed medium

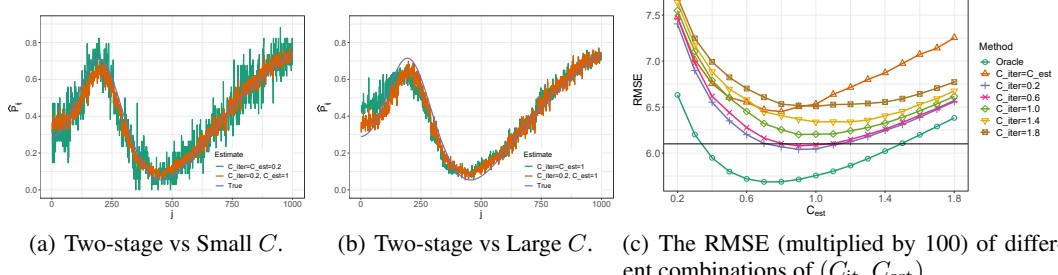

(a) Two-stage vs Small $C$.   (b) Two-stage vs Large $C$.   (c) The RMSE (multiplied by 100) of different combinations of $(C_{\text{it}}, C_{\text{est}})$.

Figure 3: *The performances on the estimation of $\mathbf{P}_{350.}$ and the whole $\mathbf{P}$ by the strategy of using a fixed $C$ and our two-stage strategy with different combinations of $(C_{\text{it}}, C_{\text{est}})$.*

size of the similar vertex set. However, according to the orange curve, if we force $C_{\text{it}} = C_{\text{est}}$, no matter what the value is, this strategy is always defeated by our two-stage strategy. The performance of the combination $(C_{\text{it}} = 0.2, C_{\text{est}} = 1)$ is the most competitive and even comparable to that of "Oracle."

The two-stage strategy makes our proposed method more data-driven, thus estimating those connecting probability matrices with complicated structure more powerfully. To demonstrate this advantage, we display the estimates with different methods on the network above in Figure 4. The estimate of USVT is not smooth overall as the low rank assumption is not satisfied. SAS also performs poorly, as the degree is non-monotone. NS recovers the smooth areas successfully but tends to over-smooth the local structure, as Zhang et al. (2017) have claimed. Our method combines the advantages of USVT and NS. It discovers the local structure while maintaining the smoothness of the whole matrix.

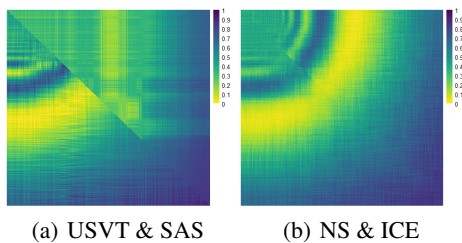

(a) USVT & SAS   (b) NS & ICE

Figure 4: *Probability matrix estimated by different methods on simulated networks . Column 1: USVT (lower) and SAS (upper). Column 2: NS (lower) and our method (upper).*

For our two-stage neighborhood selection strategy, it is important to select appropriate tuning parameters. We apply the grid search method based on the network cross-validation proposed by Li et al. (2020). As Algorithm 2 displays, the edges are split into the training set $E_{\text{train}}$ and the validation set $E_{\text{val}}$. With estimate $\widehat{\mathbf{P}}$ obtained from the training network $G(V, E_{\text{train}})$, the error on the validation set can be evaluated with negative log-likelihood loss, which is defined as $L = -\sum_{(i,j) \in E_{\text{val}}} \left[ A_{ij} \log \widehat{P}_{ij} + (1 - A_{ij}) \log \left( 1 - \widehat{P}_{ij} \right) \right]$. The black horizontal line in Figure 3(c) presents the RMSE with selected parameters, which shows the reliability of the cross-validation.

## 4   Theoretical Properties

We study the theoretical properties of our iterative method. All of the results below are based on the following assumptions about the network structure.

**Assumption 1** *Let $0 = z_0 < z_1 < \cdots < z_K = 1$, $I_k = [z_{k-1}, z_k)$ for $1 \leq k \leq K - 1$ and $I_K = [z_{K-1}, z_K]$. Assume that the graphon function $f : [0, 1]^2 \to [0, 1]$ is bi-Lipschitz on each*

**Algorithm 2** Tuning parameters selection of ICE via edge cross-validation

---

**Input:** observed adjacency matrix $\mathbf{A}$; the training proportion $p$; the candidate set $T$ of the tuning parameters $(C_{\mathrm{it}}, C_{\mathrm{est}})$; the number of replications $M$.

**Output:** a combination of the tuning parameters $(\widehat{C}_{\mathrm{it}}, \widehat{C}_{\mathrm{est}})$.

1: **for** $m = 1, \ldots, M$ **do**
2:      Randomly sample a subset of edges from $E$ with probability $p$ to obtain the training set of the edges $E_{\mathrm{train}}$. Let $E_{\mathrm{val}} = E - E_{\mathrm{train}}$ denote the validation set.
3:      Apply matrix completion method to the adjacency matrix corresponding to the leftover training network $G(V, E_{\mathrm{train}})$ to obtain $\mathbf{A}_{\mathrm{train}}$.
4:      Apply ICE method on $\mathbf{A}_{\mathrm{train}}$ to estimate $\mathbf{P}$ with each $(C_{\mathrm{it}}, C_{\mathrm{est}})$, and calculate the negative log-likelihood loss $L_m(C_{\mathrm{it}}, C_{\mathrm{est}})$ of the estimate on $E_{\mathrm{val}}$.
5: **end for**
6: Let $L(C_{\mathrm{it}}, C_{\mathrm{est}}) = \sum_{m=1}^{M} L_m(C_{\mathrm{it}}, C_{\mathrm{est}})/M$.
7: **return** $(\widehat{C}_{\mathrm{it}}, \widehat{C}_{\mathrm{est}}) = \arg\min_{(C_{\mathrm{it}}, C_{\mathrm{est}}) \in T} L(C_{\mathrm{it}}, C_{\mathrm{est}})$.

---

$I_k \times I_l$ for $1 \leq k, l \leq K$. That is, both $|f(x_1, y) - f(x_2, y)| \leq L|x_1 - x_2|$ and $|f(x, y_1) - f(x, y_2)| \leq L|y_1 - y_2|$ hold for all $x, x_1, x_2 \in I_k$ and $y, y_1, y_2 \in I_l$, where $L$ is a global constant.

**Assumption 2** *The number of $K$ grows with $n$ as $\min_k |I_k|/(n^{-1} \log n)^{1/2} \to \infty$, where $|I_k|$ denotes the length of $I_k$.*

For any $\xi_i \in [0, 1]$, let $I(\xi_i)$ denote the interval that includes $\xi_i$, let $\mathcal{N}_i(\Delta_n) = [\xi_i - \Delta, \xi_i + \Delta_n] \cap I(\xi_i)$ denote the neighborhood of $\xi_i$ in which $f(x, y)$ is Lipschitz in $x \in \mathcal{N}_i(\Delta_n)$ for any fixed $y$. Then according to the Lemma 1 in Zhang et al. (2017), for arbitrary global constant $C, C_1 > 0$, define $\Delta_n = \left[ C + (C_1 + 4)^{1/2} \right] (n^{-1} \log n)^{1/2}$, then with probability $1 - 2n^{-C_1/4}$, we have $\min_{i \in V} |\{i' : \xi_{i'} \in \mathcal{N}_i(\Delta_n)\}| \geq C(n \log n)^{1/2}$. That is, if $n$ is large enough, the size of the similar vertex set of each vertex is larger than $C(n \log n)^{1/2}$ with high probability. For simplicity, we let $C = 1$ as Zhang et al. (2017) recommend.

Based on the assumptions, our goal is to obtain $\widehat{\mathbf{P}}$ which minimizes $||\mathbf{P} - \widehat{\mathbf{P}}||_F$ by neighborhood averaging. If we can get the true set of similar vertices $S^*$, then based on $\mathbf{A}$, the corresponding estimate is $\widetilde{\mathbf{P}}$ with $\widetilde{P}_{ij}$ defined as (1), which serves as the "Oracle" estimate for neighborhood averaging methods. We have the following results.

**Theorem 1** *Let $C_2, C_3, C_4, C_5, C_6 > 0$ be arbitrary global constants and assume $n$ is large enough so that $n^{-2} + (C_4 + 8)^{1/2} n^{-1} \leq \left[ C_5 - (C_2 + 3)^{1/2} - (C_3 + 3)^{1/2} \right] \left( n^{-3} \log n \right)^{1/4}$ and $8L^2 \left[ 1 + (C_1 + 4)^{1/2} \right] (n^{-1} \log n) \leq (C_6 - 2C_5)(n^{-3} \log n)^{1/4}$, let $C_* = \max\{C_1/4, 2C_2/3, 2C_3/3, C_4/2\}$, then with probability $1 - 8n^{-C_*}$, we have*
$$||\widetilde{\mathbf{P}} - \mathbf{P}||_F^2 n^{-2} \leq C_6 (n^{-3} \log n)^{1/4}.$$

Theorem 1 shows the best error rate we can achieve once we succeed to find all the true similar vertices, which seems unrealistic in practice. Luckily, we can reach the same error rate even if the set of similar vertices includes a small number of non-similar vertices. We formulate this property in Theorem 2.

**Theorem 2** *With arbitrary global constants $C_7, C_8$ satisfying $C_8 > 2C_5 + 4C_7 > 0$ for the $C_5$ from Theorem 1, assume that $P_{ij} \in [a, b]$ for all $(i, j) \in V \times V$ where $0 < a < b < 1$, for any solution $(\mathbf{S}, \widehat{\mathbf{P}})$ where $\mathbf{S}$ satisfies*

$$\max_{(i,j)} \sum_{i' \in S_i} \sum_{j' \in S_j} \mathbb{I}((i', j') \notin S_i^* \times S_j^*) \leq \frac{\sqrt{C_7} n^{5/8} (\log n)^{9/8}}{b - a} = e(n), \qquad (3)$$

*if $n$ is large enough so that $16L^2 \left[ 1 + (C_1 + 4)^{1/2} \right] (n^{-1} \log n) \leq (C_8 - 4C_7 - 2C_5)(n^{-3} \log n)^{1/4}$, then with probability $1 - 8n^{-C_*}$, we have*
$$||\widehat{\mathbf{P}} - \mathbf{P}||_F^2 n^{-2} \leq C_8 (n^{-3} \log n)^{1/4}.$$

According to Theorem 2, for each pair $(i, j)$, since $|S_i \times S_j| = s^2$, the maximum error rate allowed for neighborhood selection is $e(n)/s^2 \approx e(n)/n \log n = \sqrt{C_7}(n^{-3} \log n)^{1/8}/(b - a)$, which is greater then $C_8(n^{-3} \log n)^{1/4}$. That is, we can obtain an estimate with low error rate even with relatively high error rate on neighborhood selection. The smaller $b - a$ is, the larger the error rate is allowed.

If we use the estimates with existing methods as the initial values, we can achieve a small error rate on neighborhood selection with a high probability, thus improving the estimation performance. For any estimate $\widehat{\mathbf{P}}^{(0)}$ which satisfies that $\max_{i \in V} \sum_{j=1}^{n} \left( \widehat{P}_{ij}^{(0)} - P_{ij} \right)^2 n^{-1} \leq C_9 E(n)$ with probability $1 - n^{-C_{10}}$, where the error rate $E(n)$ satisfies $\lim_{n \to \infty} E(n)/(n^{-3} \log n)^{1/4} \to \infty$, we have the following theorem for the estimate in the next iteration.

**Theorem 3** *Define the lower bound of the distance between each vertex $i$ and any vertex $i'' \notin S_i^*$ as*

$$C(n) = \min_{i \in V, i'' \notin S_i^*} |P_{ij} - P_{i''j}|,$$

*assume that $C^2(n) \geq 8L^2 \left[ 1 + (C_1 + 4)^{1/2} \right]^2 (n^{-1} \log n) + 20C_9 E(n)$, then with the global constants $C_2, C_3, C_4, C_6$ from Theorem 1, when we use $\widehat{\mathbf{P}}^{(0)}$ as the initial value to obtain a new estimate $\widehat{\mathbf{P}}_{new}$, with probability $1 - 8n^{-C_*} - n^{-C_{10}}$, we have*

$$||\widehat{\mathbf{P}}_{new} - \mathbf{P}||_F^2 n^{-2} \leq C_6(n^{-3} \log n)^{1/4}.$$

The results above show that if our proposed method start from a reasonably good estimate obtained by an existing method, then it will improve upon the initial estimate iteratively, making the final estimate comparable to that with all the true similar vertices. The next question is whether we can stop the iterations when the current estimate is near $\mathbf{P}$. Assume that the current estimate $\widehat{\mathbf{P}}^{(m)}$ is smooth enough to satisfy Assumptions 1 and 2, we have the following theorem.

**Theorem 4** *With arbitrary global constant $C_{11}$ and $C_{12}$ satisfying $C_{12} > 4C_5 + 2C_{11} > 0$ for $C_5$ from Theorem 1, and let $n$ be large enough so that $32L^2 \left[ 1 + (C_1 + 4)^{1/2} \right]^2 (n^{-1} \log n) \leq (C_{12} - 4C_5 - 2C_{11}) (n^{-3} \log n)^{1/4}$, if $\widehat{\mathbf{P}}^{(m)}$ satisfies $||\widehat{\mathbf{P}}^{(m)} - \mathbf{P}||_F^2/n^2 \leq C_{11}(n^{-3} \log n)^{1/4}$, then with probability $1 - 8n^{-C_*}$, we have*

$$||\widehat{\mathbf{P}}^{(m)} - \widehat{\mathbf{P}}^{(m+1)}||_F^2 n^{-2} \leq C_{12}(n^{-3} \log n)^{1/4}.$$

For our learning task, the adjacency matrix $\mathbf{A}$ is observed while the pairwise vertex distances and the sets of similar vertices can be viewed as unobserved data, and our goal is to estimate $\mathbf{P}$ with incomplete data, which is a typical task for an EM type algorithm and motivates our iterative procedure. However, although updating $\widehat{\mathbf{P}}$ with $\mathbf{S}$ is the traditional M step, updating $\mathbf{S}$ using the current estimate $\widehat{\mathbf{P}}$ is not equivalent to the E step. Therefore, we cannot directly use the existing theory.

## 5 Experiments

### 5.1 Simulated Networks

To evaluate the effectiveness of our proposed method, we compare its performance with several popular estimation methods using simulated networks with different features, including low-rank, degree monotonicity and local structure. Each network contains 1000 vertices. The connecting probability matrices are displayed in Figure 5, where the rows and the columns are ordered by $\boldsymbol{\xi}$. Graphon 1 is low rank and has vertex degree monotonicity. Graphon 2 is roughly monotone with local structure. Graphon 3 is a periodic function. Other graphons are all full-rank with extreme values in local area like the diagonal and the corner.

We compare the following methods: universal singular value thresholding algorithm (USVT) (Chatterjee, 2015), stochastic blockmodel approximation algorithm (SBA) (Airoldi et al., 2013), sorting and smoothing method (SAS) (Chan and Airoldi, 2014), neighborhood smoothing method (NS)

(Zhang et al., 2017), our proposed method (ICE). For fair comparison, we use random neighborhood averaging estimate as the initial value. We also present the result of neighborhood averaging with true similar vertices (Oracle) as a golden standard.

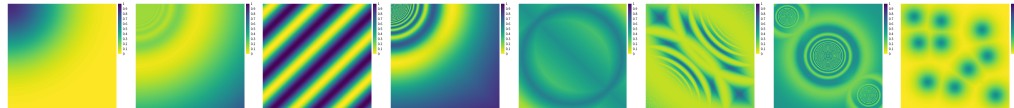

Figure 5: *Graphon functions (1-8, from left to right) to generate simulated networks.*

We show the results on average of 100 repetitions in Table 1. For each method, we report the mean of RMSE with standard deviation in the bracket. For probability matrices with low rank (Graphons 1 and 3), USVT does a good job. However, if the graphon is full rank, the low-rank approximation by USVT becomes insufficient. Although SAS method performs well on Graphons 1 and 2 since the expected degrees is monotone, it does not work on other graphon functions. NS method performs well in many cases, which coincides with the numerical results of Zhang et al. (2017). Our proposed method is comparable to NS on simple networks generated by Graphons 1 and 3. Notably, on networks with complicated structure (Graphon 4-8), our method shows significant advantage for its ability to discover local structure.

We also display the tuning parameters $(C_{it}, C_{est})$ selected via cross-validation and the best $C_{est}$ if we use true similar vertices (Oracle). The $C_{est}$ we select is comparable to that with true similar vertices while the selected $C_{it}$ is always smaller, which again shows the advantage of our method on the trade-off between the smoothness of the estimate and the ability to discover local structure.

Table 1: *The RMSE (multiplied by 100) of different methods and $C_{it}$ and $C_{est}$ selected for ICE by cross-validation on simulated networks.*

| | RMSE (multiplied by 100) | | | | | | $C_{it}$ and $C_{est}$ selected | | |
| Graphon | USVT | SBA | SAS | NS | ICE | Oracle | ICE $C_{it}$ | ICE $C_{est}$ | Oracle $C_{est}$ |
|---|---|---|---|---|---|---|---|---|---|
| 1 | **1.62(0.05)** | 4.79(0.06) | 1.72(0.04) | 2.55(0.04) | 2.23(0.18) | 1.55 | 1.05(0.40) | 1.85(0.43) | 2.33 |
| 2 | 3.74(0.03) | 38.66(0.05) | 4.92(0.06) | 3.78(0.03) | **2.87(0.05)** | 2.40 | 1.13(0.43) | 2.37(0.15) | 2.33 |
| 3 | **2.67(0.03)** | 33.19(0.46) | 34.05(0.20) | 3.33(0.03) | 3.21(0.13) | 2.49 | 0.47(0.47) | 1.43(0.33) | 1.67 |
| 4 | 7.53(0.05) | 43.55(0.03) | 14.00(0.07) | 7.25(0.05) | **6.09(0.08)** | 5.71 | 0.23(0.07) | 1.07(0.14) | 0.67 |
| 5 | 6.41(0.03) | 8.35(0.02) | 8.24(0.02) | 6.85(0.05) | **4.19(0.09)** | 3.55 | 0.58(0.16) | 2.03(0.32) | 1.78 |
| 6 | 7.22(0.03) | 12.86(0.01) | 12.74(0.01) | 6.54(0.06) | **5.20(0.10)** | 4.51 | 0.28(0.07) | 0.95(0.12) | 0.67 |
| 7 | 8.09(0.09) | 9.53(0.03) | 9.41(0.02) | 8.21(0.04) | **6.84(0.11)** | 5.93 | 0.39(0.13) | 2.27(0.37) | 1.00 |
| 8 | 4.50(0.02) | 10.61(0.03) | 10.51(0.02) | 3.88(0.05) | **3.30(0.04)** | 2.71 | 0.49(0.16) | 1.67(0.00) | 1.00 |

## 5.2 Real Networks

We analyze a human brain projectome dataset from an experiment of Beijing Normal University in China (Yan et al., 2009)[2]. The dataset is available on https://NeuroData.io/, a platform that enables large-scale neurodata storing, analyzing, and modeling. A "projectome" is a large-scale mapping between regions of the brain. This data include 48 anonymous healthy students as controls and each of them has 3 scans in different statuses. A valuable task is to the study the relationship between the regions, which may reveal the way they work together.

We have evaluated the performance of our methods on several subjects. For the sake of brevity, we only display the result on estimating the projectome of the subject with ID 0027055 in the first status[3]. The network consists of 349 vertices and 3772 edges. We present the adjacency matrix in Figure 6(a). The rows and columns are ordered according to the function of the regions.

As is shown in Figure 6, the estimates of these methods are quite different. USVT tends to estimate the probability matrix as a SBM model with many blocks. It fails to recover the local structure in the top left. SBA and SAS both perform poorly, because the degree disruption does not coincide with the connecting behaviors. The result of NS is more similar to that of ICE. They both obtain a

---

[2]http://dx.doi.org/10.15387/fcp_indi.corr.bnu1

[3]http://mrneurodata.s3-website-us-east-1.amazonaws.com/BNU3/ndmg_0-0-48/graphs/DS00350/sub-0027055_ses-1_dwi_DS00350.gpickle

smooth estimate with local structure. The difference is that, since the whole network is sparse, NS is easier to underestimate the connecting probability of moderately connected pairs. This shortcoming derives from its assumption about the smoothness, which may lead to over-smoothing.

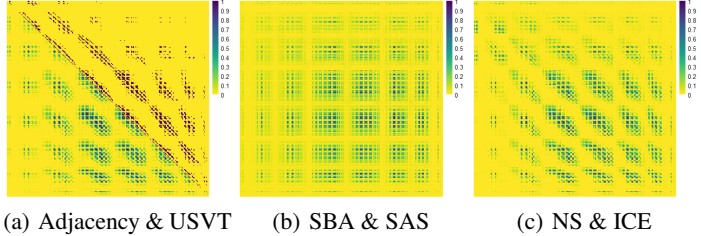

(a) Adjacency & USVT      (b) SBA & SAS      (c) NS & ICE

Figure 6: *Adjacency matrix and probability matrix estimated by different methods. Column 1: Adjacency matrix (lower) and USVT (upper). Column 2: SBA (lower) and SAS (upper). Column 3: NS (lower) and ICE (upper).*

As the true probability matrix is unknown, it is hard to examine and compare the results of different methods on real networks. Zhang et al. (2017) evaluate their method by applying it to the task of link prediction. Assume that $\mathbf{A}$ is the true adjacency matrix while $\mathbf{A}_{\text{miss}}$ is the observed version. Assume that each edge in the true network will be removed in the observed network with probability $p$, which can be formulated as $\mathbf{A}_{\text{miss}} = \mathbf{M} * \mathbf{A}$, where $M_{ij} \overset{i.i.d.}{\sim} \text{Bernoulli}(1-p)$. Based on $\mathbf{A}_{\text{miss}}$, the estimate $\widehat{\mathbf{P}}$ gives the scores proportional to the connecting probabilities (Gao et al., 2016). With threshold $t > 0$, Zhang et al. (2017) define the false positive rate and the true positive rate by

$$\text{FP}(t) = \sum_{i,j} \mathbb{I}\left(\widehat{P}_{ij} > t, A_{ij} = 0, M_{ij} = 0\right) / \sum_{i,j} \mathbb{I}\left(A_{ij} = 0, M_{ij} = 0\right),$$
$$\text{TP}(t) = \sum_{i,j} \mathbb{I}\left(\widehat{P}_{ij} > t, A_{ij} = 1, M_{ij} = 0\right) / \sum_{i,j} \mathbb{I}\left(A_{ij} = 1, M_{ij} = 0\right).$$

By varying $t$, we can obtain the ROC curve and compare the precision of different methods with AUC. The missing rate $p$ is set as $0.1$.

Here we develop a new approach for evaluation of connecting probability estimation methods in practice. Although $\mathbf{P}$ is unknown, we can estimate it by one of the existing methods and apply the result $\mathbf{P}_{\text{host}}$ as the baseline. Then we generate the adjacency matrix $\mathbf{A}_{\text{gen}}$ with $\mathbf{P}_{\text{host}}$ and apply different methods to estimate it. We refer to the method used to obtain $\mathbf{P}_{\text{host}}$ as the "host" method. The "host" is always the best due to the home-court advantage. We focus on those methods whose performances are competitive and even comparable to the "host", which demonstrates the capability of generalization.

To evaluate the precision on estimation, we display the results of the link prediction task in Figure 7(a). It is obvious that SBA and SAS are outperformed by others significantly. Our method is slightly better than NS and USVT due to its ability to recover the local structure with higher precision.

To evaluate the the capability of generalization, we use the approach we propose. For each "host", we generate the adjacency matrix for 100 repetitions and present the RMSE on average in Figure 7(b). When $\mathbf{P}_{\text{host}}$ is based on USVT, it outperforms others significantly because $\mathbf{P}_{\text{host}}$ is low rank. However, if the "host" is our proposed method, USVT performs poorly as it is hard to recover local structure. Under this setting, NS is also defeated by our method due to over-smoothing. When the "host" are SBA and SAS, NS outperforms others slightly. The reason is that both SBA and SAS output smooth estimated probability matrices and satisfy the assumption about smoothness of NS. No matter what the "host" is, our method is always competitive, which manifests its ability of generalization.

## 6 Conclusions

We propose an iterative connecting probability matrix estimation method which updates the set of similar vertices and the estimate progressively. It is intuitive, easy to implement and computationally

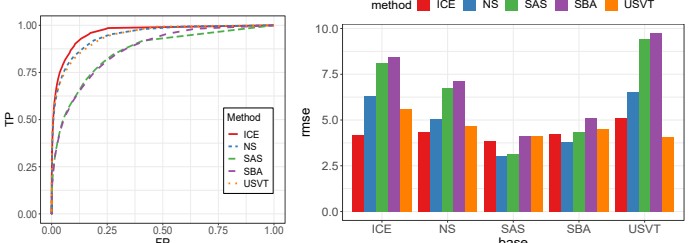

(a) *The ROC curves on the task of link prediction.*  (b) *The RMSE (multiplied by 100) with different "host".*

Figure 7: *The performances of different methods on real network.*

feasible for its fast convergence. Even starting with an estimate by random neighborhood averaging performs well. The main strength of our method is the ability to discover local and complicated structure for networks. In our algorithm, deciding the size of the similar vertex set is crucial. We propose a two-stage strategy using different sizes to balance its ability to discover local structure and output a smooth estimate. In practice, the appropriate size may vary from vertex to vertex. A possible variation of our method is to set specific size of the similar vertex set for each vertex. Although it may introduce more turning parameters, we believe that this issue is worth of further investigation.

## Acknowledgments and Disclosure of Funding

We thank the program chair and reviewers for their insightful comments that greatly improve the article. Funding in direct support of this work: None. Additional revenues related to this work: None.

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
