# OpenReview forum: "Iterative Connecting Probability Estimation for Networks"
_NeurIPS.cc/2021/Conference — NeurIPS 2021 Poster_

### Official Review · Reviewer_WMZz · 2021-07-15

**Rating:** 7
**Confidence:** 3

**Summary:**

The paper discusses a new strategy for graphon estimation that is based on an iteratively updating the graphon by considering certain neighborhoods of the nodes.


**Limitations And Societal Impact:**

There are several limitations, in particular w.r.t. the considered network model (a graphon) -- see above.

**Main Review:**

The paper discusses a novel idea for graphon estimation: instead of using a "one-shot" approach, e.g., based on a singular value decomposition, the idea is to use an procedure that --- starting from an initial guess --- updates the graphon estimate iteratively.
The problem of graphon estimation is indeed very relevant, and the presented theoretical guarantees and numerical experiments corroborate the results of the authors.

I have a number of concern regarding the paper, in terms of the presentation and the results.
First I found the initial discussion and the abstract to be somewhat misleading. The authors state that there are no strong assumptions that need to be made, but they assume that the graph is generated by a graphon model. This implies conditionally independent edges / vertex exchangability, and seems to be a pretty strong assumption to me.
I think the presentation should have been clearer in stating that the problem is that of estimating a graphon, rather than estimating the connection probability -- which is a more general (and more vague) idea.

Given that the authors concentrate on graphon models, it would have however also been useful to discuss the aspect of sparsity -- it is well known that graphons are essesntially models for dense graphs; while one can adjust the density of them to a certain extend via a scaling parameter, but obtaining truly sparse graphs would require to turn to something like a graphex, or edge-exchangable models. This is a limitation that should have been discussed in more detail in my opinion.
Indeed, we (basically) see a repercussion of this issue of graphons in Assumption 2, where the neighborhood size of a node needs to grow as (n log n)^(1/2).

I would have also liked to see more theoretical discussion (and perhaps some numerical experiments) on the choice of the C parameter; indeed as the authors point out in the conclusion, we may actually have very different C for different vertices.
Similarly, how the initial estimate of the connection probability matrix P should be chosen, and what is effect would be on the algorithm seems a question that is somewhat underdeveloped in the paper.

While some of these flaws will be very difficult to address in a revised version,  I think the paper has an interesting algorithmic idea and so I see it slight above the threshold.

=========
I thank the authors for their answers. I have updated my score, following the response and the comments from the other reviewers.

**Time Spent Reviewing:**

4

---

> ### Author Response · Authors · 2021-08-10
> **Reply to Reviewer WMZz**
>
>
> We thank the reviewer for these constructive comments.
> As the reviewer states, the proposed method updates the graphon estimate iteratively instead of using the "one-shot" estimate, and is justified both theoretically and numerically.
> We appreciate all the insightful suggestions and reply to these comments on the following aspects respectively.
>
>
> **Assumption:**
> Indeed,
> we assume that the network is generated by a graphon model, which implies conditionally independent edges and vertex exchangeability.
> The reason why we state that our method requires weak assumptions is that, compared with other graphon estimation methods which typically assume low rank or monotonicity of the degrees,
> our assumptions are relatively weak.
> As our numerical results show,
> for networks with complicated local structures, our proposed method outperforms others.
> We appreciate the feedback from the reviewer and would like to make clearer statements about our assumptions in the revision.
>
> The reason why we do not refer to our task as estimating the graphon function is that such a task sometimes can be ill-defined without strong assumptions on the network structure  (Zhang et al., 2017).
> For example, without the assumption of the monotonic expected node degree, the graphon function is not identifiable (Bickel and Chen, 2009).
> Since the main purpose of estimating the graphon function is to obtain the connecting probabilities anyway, we refer to our task as estimating the connecting probability matrix.
> And the identifiability issue of the graphon function may not matter that much in this case.
> We would like to point out the close connection between the connecting probability estimation and graphon estimation in the beginning of the manuscript to avoid confusion.
>
> **Sparsity:** The sparsity is an important feature of many real networks that we encounter in data analysis.
> One way to model sparse networks is to add a scaling parameter in the graphon model to force sparsity.
> Another way is to use truly sparse network models, such as graphex or edge-exchangeable models.
> Since our proposed method is developed under the graphon model, we expect that the method can be extended to the sparse graphon models as long as the neighborhood can be identified accurately.
> However, it is not directly applicable to graphex or edge-exchangeable models yet.
> We would love to pursue this direction in the future project, which will make the method even more practical.
>
> **Tuning Parameters:**
> The tuning parameter selection is an important and yet challenging issue.
> In the manuscript, we have developed a procedure that works well in practice.
> However, the theoretical development for the tuning parameter selection would require a thorough investigation of the two tuning parameters.
> The first tuning parameter helps us better order the vertices according to their latent positions, whereas the second tuning parameter helps us obtain better probability estimates using the correct number of vertices.
> We would like to include more discussion on the complexity of tuning parameter selection in the revised manuscript.
>
> In addition, as we discuss in the conclusion,
> a better (perhaps a more optimal) way to select the neighborhood size is to set different neighborhood sizes for different vertices depending on the shape of the graphon function.
> However, selecting so many tuning parameters simultaneously is both numerically and theoretically difficult.
> Therefore, we simplify the problem by using common tuning parameters for all the vertices in the current manuscript.
>
>
> **Initialization:**
> Due to the page limit,
> we discuss the initialization of our method in a short paragraph after Algorithm 1 in Section 3.1.
> For our method, any reasonable estimates from the existing methods can be used as the initial input.
> For example, we recommend the neighborhood smoothing by Zhang et al. (2017) as the initial estimate.
> Meanwhile, a random initial value, i.e., random selected neighborhoods, would also work.
> As Figure 2(b) in Section 3.2 shows, after sufficient number of iterations, the performance of our method with different initial values becomes comparable, which shows that our method is relatively insensitive to the initialization.
> We would like to include more discussion on the issue of initialization in the revised manuscript.

---

### Official Review · Reviewer_9KpH · 2021-07-16

**Rating:** 8
**Confidence:** 4

**Summary:**

The authors propose an iterative approach to estimate the probability of forming an edge between any two vertices in a graph, which they denote as the *connecting probability*. With only a single realization of the graph, it is a difficult task to estimate the connecting probabilities. The proposed iterative connecting probability estimation (ICE) method uses neighborhood averaging, similar to Zhang et al. (2017), but with iterative refinement of the neighborhood. They provide a theoretical analysis demonstrating that the error rate of their proposed estimation procedure is asymptotically as good as that of an "oracle" estimate that has access to the true connecting probabilities to choose the neighborhood. They also provide extensive simulations and a real data experiment demonstrating good empirical performance compared to prior work.


**Limitations And Societal Impact:**

The authors provide a very brief discussion of limitations in the conclusion. Some discussion of societal impact is provided in the supplement.


**Main Review:**

**Originality: Moderate**
The basic idea is straightforward enough that I would consider it to be low in originality: iterate a neighborhood averaging algorithm multiple times until convergence. The two-stage neighborhood selection approach using different $C$ values is innovative and appears to be effective.

I did not carefully check the proofs in the supplement, so I cannot assess if there is any originality to the analysis techniques.

**Quality: High**
The authors present a principled approach extending the neighborhood smoothing approach of Zhang et al. (2017) and present simulation results along the way justifying their decisions. These are supplemented with more extensive simulation experiments showing very impressive performance on lots of different graphon functions and a real data experiment.

They also present theoretical results that I find to be of value. It is good to see that the proposed iteration has the same asymptotic error rates as an "oracle" estimator that knows the true neighbors. Again, I did not check the proofs for correctness.

**Clarity: High**
I found the paper to be extremely well written and very easy to understand. Thank you! I have just a few minor points for improvement below.

The authors use the term "neighbors" of a vertex $i$ to denote a group of vertices with similar connecting probabilities compared to $i$. This may be confusing to some readers because the term "neighbors" is usually used to denote vertices connected to $i$. Perhaps this could be clarified with a 1-sentence explanation in the intro.

In Figure 4, showing 2 probability matrices in a single figure is confusing. Since the matrices are symmetric, only a triangular portion is necessary, which allows the matrices to be shown in this way, but there is plenty of room to show all 4 matrices as separate subfigures.

Table 1: Use bold to denote the method that performs the best in terms of RMSE (excluding oracle) to simplify comparison between methods.

Why did you choose the single test subject for the real data experiment?

**Significance: Moderate**
This paper certainly advances statistical network estimation, but it is an incremental improvement on Zhang et al. (2017), which limits significance somewhat in my view. The approach could have some broader impact given the wide applicability of network analysis in many disciplines.

**Comments after author response and discussion:**
I find the proposed method to be simple and provably effective, which is a very good combination. All the paper needs is a few minor revisions, such as changing the use of the word "neighbor", as discussed. Adding some more experimental results on the projectome data, even in the supplementary, would also strengthen the paper.


**Time Spent Reviewing:**

2 hours

---

> ### Author Response · Authors · 2021-08-10
> **Reply to Reviewer 9KpH**
>
> We thank the reviewer for the extensive and positive comments.
> As the reviewer states,
> our method extends the neighborhood smoothing approach in Zhang et al. (2017) by allowing iterative estimation so that the data can be utilized more thoroughly.
> The initial input in our method can be the results of existing methods or even a random estimate, making it easy to implement.
> We appreciate all the useful suggestions and reply to the comments on the following aspects respectively.
>
> **Originality and Significance:**
> We agree with the reviewer's assessment that the idea of iterating the neighborhood averaging estimate is straightforward,
> making it easy to understand and implement.
> Meanwhile, to the best of our knowledge, this is the first attempt in the literature of connecting probability estimation to use this two-stage neighborhood selection approach (along with the tuning parameter selection procedure).
> Such a procedure can help us balance the smoothness of the estimate and the ability to discover local structures efficiently.
>
> **Clarity:**
> The term "neighbors": Indeed, we have noticed that the use of the term “neighbors” may confuse readers.
> We would like to refer to them as "similar vertices" or "vertices with similar latent positions" to avoid any confusion.
> Meanwhile, we would like to introduce a separate definition in Section 2 to explain this concept.
> This comment is also consistent with the comments from other reviewers.
>
> Figure 4: We agree with the reviewer and will revise the figures accordingly in our final version.
>
> Table 1: Yes, we will highlight the method with the best performance in bold.
>
> Single test subject:
> We originally tested our methods on multiple subjects to verify the advantage of the proposed method.
> We found out that the results are very similar, i.e., clear advantage of the proposed method over other methods.
> Due to the space limit, we randomly select an example to display in the manuscript.
> In our revision, we would like to provide some summary statistics for all the subjects to make the comparison more comprehensive.

---

> > ### Comment · Reviewer_9KpH · 2021-09-01
> > **"Similar vertices" is better than "neighbors"**
> >
> > I think the authors' suggestion of "similar vertices" is indeed better than "neighbors". This sentence in the paper can be modified:
> >
> > Assume that there is such a group of vertices who share similar connecting patterns with vertex i and we refer to them as its “neighbors”.
> >
> > to
> >
> > Assume that there is such a group of vertices who share similar connecting patterns with vertex i; we refer to them in the rest of this paper simply as its "similar vertices".

---

### Official Review · Reviewer_JKKC · 2021-07-16

**Rating:** 6
**Confidence:** 2

**Summary:**

The paper studies an interactive algorithm for connecting probability estimation. The proposed algorithm updates the pairwise vertex distance, neighborhood set, and the connecting probabilities iteratively to improve the estimation until the result converges. The authors have shown both theoretical and empirical results. Some practical techniques to improve the performance of the algorithm are also discussed in the paper.

**Limitations And Societal Impact:**

Yes

**Main Review:**

I think this paper has some interesting results. The proposed algorithm is simple, efficient, and well-motivated. The experiments are well designed to demonstrate the advantages of the algorithm when compared with other approaches. The theoretical results proved reasonable guarantees for the performance of the algorithm.

I have a question about the algorithm. The proposed algorithm has an EM flavor. A natural question for such an algorithm is whether or not the algorithm is guaranteed to terminate and how many steps it takes to converge. Although some relevant analysis is given, I would appreciate a discussion for the general case, whether the result is positive or not.

**Time Spent Reviewing:**

6 hrs

---

> ### Author Response · Authors · 2021-08-10
> **Reply to Reviewer JKKC**
>
> We thank the reviewer for these insightful comments.
> Indeed, our iterative procedure has a strong EM algorithm flavor.
> The adjacency matrix can be considered as the observed data,
> while the latent positions (and neighborhood sets) can be viewed as the missing/unobserved data.
> The estimation of the connecting probability matrix can be conceptually carried out by maximizing the likelihood function of the adjacency matrix with respect to the probability matrix (i.e., MLE) under some smoothness constraints of the probability matrix.
> (It is important to note that we need the smoothness constraints, otherwise the MLE becomes the observed adjacency matrix.)
> If we could transform our method into an EM algorithm,
> then we could establish more theoretical results about the convergence using the existing EM algorithm theory, such as Wu (1983) "On the Convergence Properties of the EM Algorithm," Ann. Stat.
>
> However, the proposed method and EM algorithm are slightly different.
> For example, although updating the neighborhood set $\mathbf{S}$ using the current probability matrix estimate $\widehat{\mathbf{P}}$ is equivalent to the E step,
> updating $\widehat{\mathbf{P}}$ using the current $\mathbf{S}$ is slightly different from the traditional M step.
> This is because we update each connecting probability by averaging over several vertex pairs whose connecting probabilities are similar, but not exactly the same.
> So this is not equivalent to maximizing the complete data likelihood function.
> Under the finite sample size, because of this distinction, we cannot directly use the existing theory.
>
> On the other hand, under the smoothness assumptions, as the sample size increases, the neighborhood size relative to the total sample size decreases, so the difference in connecting probabilities within the neighborhood set also vanishes.
> Therefore, we conjecture that the two methods may be asymptotically equivalent.
> If that is the case, then updating the connecting probability matrix estimate may guarantee the monotonic increase in the incomplete data likelihood function, which is one of the key features of EM to ensure convergence.
> However, to establish such an equivalence would require a thorough analysis of the incomplete and complete data likelihood functions, and therefore deserves a separate investigation.
> We would love to add more discussion on the proposed method's connection to EM algorithm and its convergence in the later version of the manuscript.

---

### Official Review · Reviewer_7aba · 2021-07-17

**Rating:** 7
**Confidence:** 3

**Summary:**

This paper deals with the task of estimating the connection probabilities in a random graph, assuming that it is drawn from a graphon model; i.e. $P_{ij} = f(\xi_i, \xi_j)$, where $f$ is a smooth function and the $\xi_i$ are uniform random variables. The authors do so by exploiting the same ansatz as in Zhang et al: for $i \in [n]$, find several $i'$ with connection probabilities "close" to those of $i$, and average the $P_{i'j}$ to find an estimate for $P_{ij}$.

The main difference with the existing litterature is the choice of the $i'$: the authors choose an iterative procedure, building ever improving estimators of the $P_{ij}$ and using the current estimate to find the $i'$ and build the next estimator. The paper also presents a two-step procedure to choose the right number of "close" vertices.

The authors then show several theoretical properties of their algorithm, in a model including (but not limited to) the classical stochastic block model. Their bounds depend on the number of vertices correctly classified as "close" to each vertex in the graph.

Finally, they present experiments on both synthetic and real-world data; these seem to indicate that the ICE algorithm outperforms its counterparts when the graphon function $f$ becomes more complex.

**Limitations And Societal Impact:**

Yes

**Main Review:**

This paper presents a novel and effective method for probability estimation in graphon models. The authors show some convergence bounds on a restricted model (functions that are Lipschitz on rectangles), but the numerical experiments seem to indicate that the algorithm converges for a more general class of functions.

Overall, the paper is well-written and easy to read, with a few caveats:
* the use of "neighbors" to denote similar vertices clashes with the overwhelming acception of this term in graph theory, which causes some confusion,
* the statements of Theorems 1-4 are very convoluted, involving up to 12 different constants; I'd rather suggest very slightly weaker but more readable versions (e.g. "fusing" some constants) in the main text, and if needed putting the exact bounds in the appendix.

Those drawbacks being fairly straightforward to fix, they do not impact the overall quality of the paper too much.

**Time Spent Reviewing:**

2

---

> ### Author Response · Authors · 2021-08-10
> **Reply to Reviewer 7aba**
>
> We thank the reviewer for these constructive comments.
> As the reviewer states, the theoretical properties of our proposed method are established under the assumptions on certain levels of smoothness of the graphon function.
> And the method also performs well on other networks generated by more complex graphon functions, according to our numerical results.
>
> We also agree with the reviewer about these suggestions.
> First, the term "neighbors" may be confusing to readers, since it may be misunderstood as the vertices that connect to other vertices.
> This suggestion is consistent with the feedback from other reviewers.
> We would like to revise this term and also introduce a definition explaining this concept in Section 2.
> For example, we can change it to "similar vertices" or "vertices with similar latent positions" and clarify this concept throughout the manuscript.
> As for the statements in our theorems, we agree with the reviewer that the use of many constants is not elegant.
> Following the reviewer's suggestions, we will try our best to fuse these constants and simplify some statements without compromising the content.

---

### Decision · Program_Chairs · 2021-09-27

**Decision:**

Accept (Poster)

**Comment:**

The paper presents a simple iterative and provably effective method for graphon estimation, which improves on previous "single shot" estimation techniques. The reviewers were in consensus that the idea is very natural and well-studied, and a significant contribution to the graphon literature.